# Integrative Study of the Life Cycle in the Marine Protist *Thraustochytrium aureum* ssp. *strugatskii*

**DOI:** 10.3390/ijms262311302

**Published:** 2025-11-22

**Authors:** Alexey V. Doroshkov, Ludmila G. Naumenko, Daniil A. Iukhtanov, Ksenia N. Morozova, Elena V. Kiseleva, Aleksei G. Menzorov, Ulyana S. Zubairova

**Affiliations:** 1The Federal Research Center Institute of Cytology and Genetics, Siberian Branch of the Russian Academy of Sciences, 630090 Novosibirsk, Russia; l.naumenko@alumni.nsu.ru (L.G.N.); iukhtanov.daniil@gmail.com (D.A.I.); morozko@bionet.nsc.ru (K.N.M.); elka@bionet.nsc.ru (E.V.K.); menzorov@bionet.nsc.ru (A.G.M.); ulyanochka@bionet.nsc.ru (U.S.Z.); 2Institute of Fundamental Biology and Biotechnology, Siberian Federal University, 660036 Krasnoyarsk, Russia; 3Department of Natural Sciences, Novosibirsk State University, 630090 Novosibirsk, Russia

**Keywords:** *Thraustochytrium aureum* ssp. *strugatskii*, Labyrinthulea, life cycle, multimodal microscopy, ultrastructure, morphometric analysis, karyokinesis, population dynamics, biotechnology, marine protists

## Abstract

*Thraustochytrium aureum* ssp. *strugatskii*, a marine protist belonging to the class Labyrinthulea, exhibits a complex life cycle characterized by alternating motile and vegetative phases. Using an integrative multimodal microscopy approach, we reconstructed its full developmental cycle and analyzed the coordination between cellular morphology, subcellular architecture, and population-level behavior. Transmission and scanning electron microscopy, combined with fluorescence and time-lapse imaging, revealed the dynamics of nuclear division, organelle rearrangement, and zoospore formation. Morphometric analysis of serial ultrathin sections demonstrated distinct changes in mitochondrial distribution, Golgi apparatus, and lipid droplet abundance during transitions between stages. We have shown that vegetative cells undergo synchronized karyokinesis coupled with stable nuclear-to-cytoplasmic ratios, leading to the emergence of multinucleate stages prior to zoospore formation. The integration of ultrastructural and dynamic data enabled us to propose a systems-level model linking metabolic state, morphogenesis, and population structure. This model highlights feedback regulation between nutrient availability, biomass accumulation, and developmental synchronization. Our results establish that *T. aureum* ssp. *strugatskii* has good potential to serve as a tractable model organism for systems-level studies of protists and provide an initial framework for predictive modeling of its life cycle under controlled conditions.

## 1. Introduction

The class Labyrinthulea, a group of heterotrophic protists belonging to the SAR supergroup, represents an ecologically important component of marine ecosystems. These organisms play an important role in the nutrition of many planktonic species and are also considered promising candidates for biotechnological applications, particularly in the production of bioactive compounds [1]. Among them, members of the order Thraustochytrida are of special interest due to their ability to accumulate lipids that can constitute up to 60% of the cell dry mass under specific conditions [2]. This exceptional lipid productivity, especially the synthesis of polyunsaturated fatty acids, has made thraustochytrids an attractive resource for industrial biotechnology.

The ongoing research efforts around the world focus on optimizing cultivation conditions and improving the yield of valuable fatty acids [3,4]. Improving the efficiency and scalability of thraustochytrid cultivation requires the development of optimized media formulations and bioprocess strategies [5]. For instance, the biomass of *Thraustochytrium aureum* has been shown to grow most efficiently in media containing yeast extract, peptone, and glucose [6], suggesting that medium composition can be fine-tuned to maximize biomass or lipid yield. However, the relationship between growth medium composition and the concentration of specific target compounds, such as omega-3 fatty acids, may not always be straightforward. Consequently, mathematical and computational modeling approaches are increasingly employed to predict biomass accumulation and the synthesis of key metabolites [1].

Recent advances in machine learning have enabled the use of experimental growth data to optimize culture performance in bioreactors [7]. At a deeper system level, genome-scale metabolic models (GEMs) integrate information on the complete biochemical pathways of the organism, allowing the simulation of metabolic behavior under different environmental and nutritional conditions [8]. Such models have proven useful for designing strategies to improve culture performance and engineer thraustochytrid strains with higher yields of target products, including docosahexaenoic acid [9].

However, most existing models rely on a limited number of input parameters and do not explicitly account for the multi-level organization of living systems, particularly the dynamic structure of cell populations [1]. For instance, it has been demonstrated that during different developmental stages, thraustochytrid cells vary markedly in both the size and the number of lipid droplets [10]. Consequently, the age composition of the population may have a substantial impact on the overall yield of the target metabolites. Therefore, the total lipid content of a culture may depend not only on intracellular metabolic regulation but also on the relative abundance of specific developmental stages within the population. This suggests that selective enrichment of cultures for particular stages of the life cycle could represent a straightforward and efficient strategy to enhance the production of desired metabolites.

To achieve a comprehensive understanding and effective control of processes occurring in bioreactors, a systems-level approach is required that integrates data across multiple biological scales. Incorporating not only biochemical and molecular-genetic parameters but also the cellular level of organization may enable more accurate modeling of culture dynamics and improve the prediction of productivity under variable environmental conditions. Such integrative frameworks can bridge the gap between molecular regulation and population-level behavior, ultimately advancing both biotechnological process optimization and the modeling of broader ecological cycles.

The isolate *Thraustochytrium aureum* ssp. *strugatskii*, according to preliminary analyses conducted by our research group [1,11], possesses the enzymatic components of the omega-3 fatty acid biosynthesis pathway, including those responsible for the synthesis of docosahexaenoic and eicosapentaenoic acids. This organism therefore holds considerable potential for the industrial production of polyunsaturated fatty acids, as well as for use as a model organism to study the ecological roles of protists, the evolution of multiple fission mechanisms, and the biosynthetic pathways of diverse metabolites. The life cycle of this protist, similar to other representatives of the order Thraustochytrida, comprises a series of distinct developmental stages. The vegetative phase begins when a motile zoospore attaches to a substrate. At this stage, the zoospore loses its flagellum and develops a bothrosome [12], an organelle responsible for the production of the ectoplasmic reticulum, a specialized membrane network of tubules that extends into the surrounding environment. Following attachment, the thraustochytrid cell grows and subsequently divides either by binary fission or through the release of newly formed zoospores, the number of which can vary widely [13,14].

Thus, in *T. aureum* ssp. *strugatskii* cultures, two main life-cycle phases can be observed: a motile stage and a sessile stage [11]. The duration of each phase and the number of daughter zoospores are variable, although the underlying regulatory mechanisms remain unknown. Notably, multinucleate stages that precede zoospore formation and the nonlinear dynamics of cell growth are characteristic of many members of the class Labyrinthulea. The isolate *T. aureum* ssp. *strugatskii* may therefore serve as a convenient model organism for studying these processes. Determining its growth parameters could help elucidate how the stability of the nuclear-to-cytoplasmic ratio is maintained across cells with different number of nuclei, and how precise cytokinesis occurs during the generation of multiple daughter cells. Due to its well-defined culture conditions and rapid growth, *T. aureum* represents an experimentally accessible model to investigate the mechanisms of multinucleate development and coordinated cell division.

In this study, we present a systems-level framework for protist biology using the isolate *T. aureum* ssp. *strugatskii*. Expanding on our earlier study [11], we now present a more detailed and integrative analysis of the life cycle organization of this organism. We combine morphological and ultrastructural data from distinct developmental stages obtained by transmission and scanning electron microscopy, fluorescence imaging, and phase-contrast microscopy, together with quantitative measurements of single-cell growth and population dynamics. This comprehensive approach links subcellular processes to population-level behavior and provides a foundation for predictive modeling of culture development, with potential applications in optimizing growth conditions in bioreactors.

## 2. Results and Discussion

### 2.1. Morphology and Life-Cycle Stages of *T. aureum* ssp. *strugatskii*

In laboratory culture, *T. aureum* ssp. *strugatskii* follows a sequence of developmental stages that together constitute its life cycle. These stages progress from (i) motile biflagellate zoospores to (ii) attached vegetative cells, (iii) enlarged multinucleated stages, (iv) cytokinesis with formation of undispersed zoospores, and finally (v) release of new free-swimming biflagellate zoospores. This sequence is summarized schematically in Figure 1, providing a visual reference for the detailed observations described below.

Motile and non-motile spherical cells were consistently observed in culture, progressing into enlarged globose structures containing internally developing zoospores. These observations define a continuous sequence of proliferative states under our conditions. To delineate these stages with high resolution, we employed a multimodal microscopy approach. Phase-contrast and fluorescence imaging were used to identify general morphology, nuclear dynamics, and stage transitions. SEM resolved the surface architecture, including the ectoplasmic network and scale layers, whereas TEM provided ultrastructural information on nuclei, mitochondria, paranuclear bodies, bothrosomes, and lipid droplets. Together, these complementary methods enabled a comprehensive reconstruction of the life-cycle progression and its underlying cellular organization.

The developmental stages observed closely resemble those reported for other *Thraustochytrium* species, highlighting a conserved proliferative strategy among thraustochytrids. The morphological features and life-cycle patterns observed in *T. aureum* ssp. *strugatskii* are largely consistent with previous descriptions of other Labyrinthulea members [15,16,17].

#### 2.1.1. Vegetative Cell

Like other thraustochytrids, *T. aureum* ssp. *strugatskii* spends most of its life cycle in the attached phase, represented by the vegetative cell (Figure 1). During growth, these cells increase in diameter from approximately 2 µm to more than 10 µm. In laboratory culture, vegetative cells exhibit a rounded morphology and considerable size variation, indicating that cytokinesis occurs asynchronously among individual cells. Small vegetative cells are shown in Figure 2A,B, illustrating the characteristic shape and the development of an extensive ectoplasmic network attached to the substrate.

Transmission electron microscopy (TEM) revealed that the cytoplasm of vegetative cells is densely packed with ribosomes and contains all major eukaryotic organelles. The nucleus is typically rounded and centrally located, whereas mitochondria are distributed throughout the cytoplasm and display tubular cristae (Figure 2C,D and Appendix A). The Golgi apparatus consists of compact stacks of flattened cisternae. The number and size of lipid droplets vary markedly among cells: some contain several large inclusions (Appendix A), whereas others lack them entirely (Figure 2C–E and Appendix A). This heterogeneity, together with differences in the number of ectoplasmic branches, contributes to the pronounced morphological diversity within the population.

In addition to common eukaryotic features, vegetative cells of *T. aureum* ssp. *strugatskii* possess several structures characteristic of Labyrinthulea, including the ectoplasmic network, bothrosome, surface scales, and the paranuclear body. The ectoplasmic network originates from the bothrosome, which is an electron-dense organelle composed of compact folds of smooth endoplasmic reticulum (Figure 2C,D and Appendix A). It serves as a conduit for extracellular material exchange and adhesion, extending beyond the cell surface and forming overlapping branched threads that often interconnect with those of neighboring cells. As a result, individual network territories are difficult to distinguish.

TEM sections show rounded profiles corresponding to cross sections of the ectoplasmic network or vesicular secretions between surface scales (Figure 2D,E). In some cells, mitochondria and lipid droplets occur in close proximity or direct contact (Appendix A), suggesting transient metabolic interactions between these organelles (Figure 2D). Such contacts, sometimes described as metabolic synapses, may facilitate lipid transfer to mitochondria and support energy metabolism during rapid growth.

Another distinctive feature of the vegetative cell is the paranuclear body, a ribosome-free cytoplasmic region containing smooth endoplasmic reticulum with expanded cisternae (Figure 2F). Its consistent spatial association with the nucleus and mitochondria suggests a role in intracellular transport and organelle communication, possibly coordinating nuclear and mitochondrial metabolism during active growth.

Taken together, these observations indicate that the vegetative cell of *T. aureum* ssp. *strugatskii* possesses a complex cellular architecture incorporating both typical eukaryotic components and unique labyrinthulean features. The ectoplasmic network derived from the bothrosome probably mediates nutrient acquisition and substrate attachment during saprotrophic growth, while dynamic interactions among mitochondria, lipid droplets, and the endoplasmic reticulum likely reflect metabolic reorganization associated with energy transitions and the onset of the next life-cycle stages.

#### 2.1.2. Multinucleated Stages and Cytokinesis Patterns

The observations described above indicate that vegetative cell division in *T. aureum* ssp. *strugatskii* may occur after any round of karyokinesis. Therefore, the stage previously referred to by other authors as the “zoosporangium” is more precisely interpreted as a multinucleated vegetative cell in which cytokinesis has been initiated. To investigate this process, we examined the morphology of dividing vegetative cells using serial-section transmission electron microscopy (TEM) and three-dimensional reconstruction.

During the late vegetative stage, nuclei were positioned near the cell periphery, suggesting a preparatory redistribution preceding cytokinesis. A three-dimensional model of an eight-nucleate cell (Figure 3A) shows evenly distributed nuclei and nascent zoospore primordia surrounding a central region of free cytoplasm. When the number of nuclei exceeded sixteen, daughter cell precursors were arranged mainly along the cell periphery, forming a central residual cytoplasmic region devoid of nuclei (Figure 3C,D). This organization is reminiscent of multinucleate division in apicomplexans [18], where each nucleus occupies a distinct territory of the plasma membrane while a residual cytoplasmic remnant remains after cytokinesis.

Cells at the multinucleated stage reached diameters of 6–10 µm or more (Figure 3A). Vegetative cells gradually increased in both size and nuclear number through successive rounds of karyokinesis, eventually releasing zoospores upon completion of cytokinesis. Based on morphological and ultrastructural evidence, no distinct structural or biochemical marker separates the so-called zoosporangium from a large multinucleated vegetative cell. Their main differences lie in overall size and in the degree of ectoplasmic network development, which varies depending on culture conditions. For this reason, we interpret the zoosporangium not as an independent stage of the life cycle but as a mature multinucleated vegetative cell approaching cytokinesis.

To further characterize the intracellular organization of this stage, we reconstructed a large multinucleated cell from serial ultrathin TEM sections (Figure 3A–D; Appendix A). The same major eukaryotic organelles observed in the single-nucleated vegetative stage were also present. Mitochondria formed a branching tubular network, suggesting that each cell may contain a continuous mitochondrial reticulum. Nuclei were positioned close to the plasma membrane, and the Golgi complex with stacked cisternae was localized near the nuclear envelope. The paranuclear bodies were interconnected, forming an extended network that occupied much of the central cytoplasmic volume and established contacts with lysosomes, lipid droplets, and mitochondria (Figure 3E). Centrioles were typically located adjacent to the nuclei.

Although the ectoplasmic network was absent at this stage, the bothrosome remained present and appeared as a smooth endoplasmic reticulum-derived stack containing electron-dense material (Figure 3F). Some cells retained surface scales, whereas others lacked an external membranous layer, possibly reflecting intermediate remodeling preceding zoospore release.

Taken together, these observations indicate that cytokinesis in *T. aureum* ssp. *strugatskii* involves coordinated segregation of nuclei along the cell periphery, followed by membrane partitioning and cytoplasmic condensation into distinct daughter cells. The persistence of a paranuclear network and extensive interactions among mitochondria, endoplasmic reticulum, and lipid droplets suggest that organelle rearrangement plays a central role in the transition from multinucleated vegetative stages to zoospore formation.

#### 2.1.3. Zoospore Formation

The vegetative phase of *T. aureum* ssp. *strugatskii* gives rise to multinucleated cells that subsequently differentiate into zoospores. This process begins with extensive nuclear proliferation within the cytoplasmic mass, accompanied by an increase in both cell size and cytoplasmic density. During cytokinesis, nuclei become evenly distributed beneath the plasma membrane, each defining a separate cytoplasmic domain that later forms an individual zoospore. At this stage, the outer membrane remains continuous, enclosing all developing daughter cells within a common envelope (Figure 4A–C).

Scanning electron microscopy revealed groups of newly formed zoospores still connected by remnants of the parental membrane (Figure 4A,B). Confocal laser scanning microscopy confirmed the spatial arrangement of nuclei and cytoplasmic boundaries during cytokinesis. Early stages show nascent zoospores located beneath the intact plasma membrane, with a central region devoid of cytoplasmic material (Figure 4C). At later stages, clusters of mature zoospores surround a residual body of undivided cytoplasm (Figure 4D). This pattern suggests a mechanism involving surface-associated nuclear territories, similar to those observed in other multinucleated protists, where membrane invagination proceeds radially around each nucleus before complete cell separation.

After cytokinesis, undispersed zoospores remain enclosed within the parental envelope, forming compact clusters of 4–16 daughter cells depending on the number of nuclear divisions. These undispersed zoospores exhibit tightly packed fully differentiated internal structures, including a central nucleus, mitochondria, Golgi stacks, a paranuclear body, and several lipid droplets (Figure 4E,F and Appendix A). Mitochondria form interconnected tubular networks, while the Golgi complex is closely associated with the nucleus. The paranuclear body frequently contacts multiple organelles, suggesting a role in intracellular coordination. At this stage, the flagellar apparatus is already formed, although the bothrosome and ectoplasmic network are still absent.

#### 2.1.4. Zoospore Release and Ultrastructure

Free-swimming zoospores are released following rupture or gradual cleavage of the outer cell wall (Figure 5A,B). These cells are spherical to slightly ovoid, measuring 2.5–3 µm in diameter, and bear one or two subapical flagella that provide motility. Their cytoplasm contains a central nucleus, mitochondria with tubular cristae, a Golgi complex, a paranuclear body, and small lipid droplets (Figure 5C). A characteristic feature of this stage is the presence of parallel lamellae of rough endoplasmic reticulum arranged adjacent to the nuclear envelope (Figure 5D). These rough endoplasmic reticulum structures extend from the paranuclear body and likely participate in bothrosome formation during the transition back to the vegetative stage, when rough endoplasmic reticulum membranes gradually transform into smooth endoplasmic reticulum (Figure 5E). Flagellar roots and kinetosomes were frequently observed near nascent bothrosomes, indicating spatial coupling between flagellar disassembly and ectoplasmic network regeneration.

The process of zoospore formation and release in *T. aureum* ssp. *strugatskii* therefore represents a continuous sequence of coordinated events: karyokinesis, cytoplasmic partitioning, flagellar assembly, and eventual liberation of motile cells. Despite the absence of a morphologically distinct “zoosporangium” stage, the multinucleated vegetative cell functions as a reproductive module capable of adjusting reproductive output in response to environmental conditions.

The first electron micrographs of thraustochytrid zoospores were published by Kazama in 1973 [19]. Due to the limitations of fixation and sample preparation at that time, the paranuclear body was not observed; this structure was characterized only in later studies. Kazama described a cytoplasmic compartment termed the “cytolysome,” consisting of a portion of the cytoplasm surrounded by acid phosphatase–positive vesicles and lamellae. He proposed that these vesicles and lamellae fused around a cytoplasmic region to form a closed, double-membrane compartment. We suggest that the membrane structures described by Kazama most likely correspond to the paranuclear body, which represents a ribosome-free area filled with smooth endoplasmic reticulum containing expanded cisternae. A close spatial association of the paranuclear body with mitochondria has also been reported for *Schizochytrium aggregatum* and other Labyrinthulomycetes species [12,20].

#### 2.1.5. Morphometric Analysis of Organelles

Morphometric analysis of organelle areas and their contacts was performed for cells of *T. aureum* ssp. *strugatskii* at different life cycle stages, including vegetative cells, zoosporangia, undispersed zoospores, and biflagellate zoospores. Quantified parameters included the absolute and relative areas of the cell, nucleus, mitochondria, Golgi complex, lipid droplets, and the paranuclear body (Appendix A).

Absolute areas of the nucleus, mitochondria, Golgi complex, and paranuclear body increased from the vegetative to the zoosporangium stage, followed by a decrease in both undispersed and motile zoospores. Similarly, total cell area expanded during vegetative growth, reached a maximum in the zoosporangium, and declined to near-vegetative levels in zoospores. The total lipid droplet content remained stable between vegetative and zoosporangium stages but declined sharply in undispersed and motile zoospores (Appendix A). In relative terms, the nucleus occupied a larger fraction of cell volume in zoospores, whereas the paranuclear body reached its highest proportion at the zoosporangium stage. The Golgi complex showed an increased relative area during active growth and differentiation, whereas lipid droplets dominated in vegetative cells, reflecting their high biosynthetic activity.

Multiple types of organelle contacts were identified in all stages of the life cycle, including nucleus-paranuclear body, nucleus-lipid droplet, mitochondrion-paranuclear body, mitochondrion-lipid droplet, paranuclear body-lipid droplet, and paranuclear body-mitochondrion associations (Appendix A). Most contact lengths did not differ significantly between stages, except in motile zoospores, where mitochondria-lipid droplet contacts were significantly shorter compared to undispersed zoospores and zoosporangia (p<0.05). Representative examples of lipid droplet fusion with mitochondria and their joint contacts with the paranuclear body are shown in Appendix A. These structures likely reflect shifts in cellular energy metabolism during transitions between growth and dispersal phases.

The observed increase in paranuclear body area at the zoosporangium stage supports the hypothesis that this structure is involved in regulating metabolic activity, possibly coordinating energy flux between mitochondria and lipid droplets during zoospore formation. Conversely, the reduction of lipid droplets at later stages is consistent with their utilization as an energy source during motile, non-feeding phases of the life cycle. Together, these observations suggest that organelle interactions in *T. aureum* ssp. *strugatskii* form a functional metabolic network integrating lipid metabolism, energy conversion, and intracellular transport.

Although TEM has previously revealed organelle proximities in other thraustochytrids [17], there have been no systematic descriptions of membrane contact sites in this taxonomic group. Our results provide the first qualitative and quantitative characterization of interorganelle contacts in thraustochytrids, demonstrating their persistence across all developmental stages observed under laboratory conditions. These findings highlight the presence of an integrated system of metabolic connections, particularly between lipid droplets, paranuclear body, and mitochondria, likely reflecting direct transport mechanisms and energy coordination. Such lipid-organelle interactions are widespread among stramenopiles and play a crucial role in cellular metabolism and organelle organization [21].

### 2.2. Quantitative Analysis of Growth Dynamics

#### 2.2.1. Correlation Between Cell Growth and Nuclear Multiplication

Vegetative cells of *T. aureum* ssp. *strugatskii* increase in size from the moment of attachment to the release of zoospores, the number of which varies among individual cells as in other thraustochytrids [22]. To examine the relationship between cytoplasmic growth and nuclear replication, we reconstructed three-dimensional models of 68 cells using confocal fluorescence microscopy and measured their total volume and the number and volume of nuclei within each cell. Representative optical sections at different stages of karyokinesis and the corresponding scatterplot of cell volume versus nuclear number are shown in Figure 6.

The analysis revealed a strong positive correlation between cell volume and nuclear number (Pearson’s r=0.95, p<0.05). The distributions of cell volumes associated with different nuclear counts occupy compact, partially overlapping regions, and their medians increase monotonically with nuclear number. Pairwise comparisons showed significant differences between most groups (p<0.05, *t*-test), except for the pairs 2–4, 4–8, and 32–64 nuclei. Cells containing $2^{n}$ nuclei predominated in the population, indicating synchronous karyokinesis across nuclei within a common cytoplasm. Three-dimensional fluorescence reconstructions occasionally showed granular, disk-like chromatin regions corresponding to metaphase plates and paired discs characteristic of anaphase (Appendix A). These observations confirm that all nuclei divide simultaneously, maintaining a constant nuclear-to-cytoplasmic ratio throughout vegetative growth.

These results demonstrate a close coordination between cytoplasmic growth and nuclear replication in *T. aureum* ssp. *strugatskii*. Synchronous nuclear division suggests that internal regulatory mechanisms maintain homeostasis of the nuclear–cytoplasmic ratio in multinucleated cells, consistent with observations for other thraustochytrids [23]. In contrast, this behavior differs from the asynchronous nuclear division observed in apicomplexan parasites such as *Plasmodium falciparum* [24], highlighting distinct evolutionary strategies within the SAR group.

#### 2.2.2. Population Growth Dynamics Under Different Nutrient Conditions

To evaluate how nutrient composition and concentration affect population-level growth, we monitored biomass accumulation as optical density (OD) over six days in four culture media: FAND, FAND supplemented with glucose (FANDg), YT, and YT supplemented with glucose (YTg). Each medium was tested in undiluted form and at 1:4 and 1:8 dilutions, with three biological replicates per condition. Daily OD measurements were obtained using a CLARIOstar Plus microplate reader (BMG Labtech, Ortenberg, Germany), and the results are summarized in Figure 7A.

Analysis of variance demonstrated that both the base medium composition and the addition of glucose significantly influenced biomass accumulation after three days of cultivation (p<0.05). Cultures grown in FANDg medium exhibited the highest OD values in undiluted and 1:4 conditions, whereas at 1:8 dilution, the best growth was observed in YTg medium (Figure 7A). These results indicate that carbon supplementation enhances growth in nutrient-rich conditions, while moderate dilution of complex media such as YTg can improve nutrient accessibility and promote sustained biomass accumulation.

Visual monitoring with the Cell-IQ live-cell imaging platform revealed clear morphological differences among conditions (Figure 7B). Cells grown in nutrient-rich media were larger compared with those in diluted media. In YT and YTg media, ectoplasmic network formation was generally less common but became more frequent at higher dilution (1:8), likely reflecting an adaptive increase in surface contact area and nutrient absorption efficiency under limited resource conditions.

Taken together, these data allowed us to presume that both carbon and nitrogen availability are critical determinants of biomass accumulation and morphogenetic activity in thraustochytrids, consistent with previous findings for other species [6]. Based on these results, FAND medium was selected as the standard condition for subsequent quantitative analyses of cell growth and proliferation.

#### 2.2.3. Dynamics of Vegetative Cell Growth and Division

To quantify the relationship between cell growth, division timing, and nutrient availability, we analyzed individual vegetative cells cultured in undiluted, 1:4, and 1:16 dilutions of FAND medium using the Cell-IQ live-cell monitoring system. A total of 2000 zoospores were seeded per well, and the diameter of 20 representative cells per replicate was measured from attachment to cytokinesis (Figure 8A and Appendix A, Appendix A).

In standard FAND medium, the mean duration of the vegetative phase (from attachment to division) was approximately 12.5 h, and the cell diameter reached approximately 15 µm. At a 1:4 dilution, cells divided after about 11.7 h and reached a mean diameter of 12.8 µm, whereas in the 1:16 dilution, division occurred earlier (after 9.2 h) and the maximum diameter decreased to 10 µm. Pairwise *t*-tests confirmed significant differences in final cell size between all media (p<0.05). As illustrated in Figure 8A, cell growth followed a logistic trajectory, with both slope and amplitude decreasing under nutrient limitation. Insets show representative four- and 32-nucleated cells stained with Propidium Iodide (PI, red), $4^{\prime }$,6-diamidino-2-phenylindole dihydrochloride (DAPI, blue), and Calcofluor White (CFW, blue).

In standard FAND medium, the number of daughter zoospores produced per vegetative cell increased with growth duration and correlated with nuclear number, following the characteristic 2^n^ sequence (2, 4, 8, 16, and so on), as shown in Figure 8B. A strong positive correlation was observed between the number of nuclei and the time required for division (r=0.771, p<0.001). ANOVA revealed significant differences among nuclear groups (F=18.56, p<0.001), and post hoc tests confirmed pairwise distinctions (Bonferroni-corrected *p*-values <0.01 for most comparisons). Representative 8- and 16-nucleated cells in prophase and anaphase stages (insets in Figure 8B) illustrate synchronous karyokinesis across all nuclei.

Overall, these results demonstrate that nutrient concentration directly regulates both the duration of the vegetative phase and the reproductive output of *T. aureum* ssp. *strugatskii*. Cells grown in nutrient-rich medium reached larger diameters and higher nuclear counts (up to 256), whereas under nutrient limitation they divided earlier and produced fewer zoospores (4–16). This modulation likely reflects an adaptive strategy balancing growth costs with dispersal efficiency under variable environmental conditions. The integration of morphometric, time-lapse, and population-level data indicates that cell growth, nuclear replication, and cytokinesis in *T. aureum* are tightly coordinated and environmentally responsive. The number of nuclei thus provides a reliable indicator of both cell size and reproductive potential, while medium composition determines the timing of cytokinesis and the scale of population expansion.

### 2.3. Systems Integration of Morphology, Growth, and Metabolic Organization

The life cycle of *T. aureum* ssp. *strugatskii* represents a dynamic, self-regulating system in which cellular morphology, population structure, and metabolic organization are tightly interconnected. As illustrated in Figure 9A, the integration of multimodal microscopy with quantitative morphometric analysis provides complementary information across multiple biological scales, from the ultrastructure of individual organelles to single-cell dynamics and population-level behavior. The synthesis of these datasets reveals that transitions between vegetative growth, multinucleated stages, and zoospore release are governed by coordinated morphological and metabolic reorganization (Figure 9B).

At the cellular level, size control and nuclear multiplication are synchronized through feedback mechanisms that couple cytoplasmic volume with DNA replication. This coupling maintains homeostasis in the nuclear-to-cytoplasmic ratio despite variation in nuclear number. As vegetative cells approach a threshold size, the redistribution of organelles, particularly mitochondria, Golgi apparatus, and lipid droplets, marks the onset of differentiation. These rearrangements coincide with a reduction in inter-organelle contact networks, which indicates a reallocation of metabolic flux from biosynthetic processes to energy-intensive activities associated with cytokinesis and motility.

At the population level, these cellular processes manifest as collective oscillations between growth and division phases. The relative abundance of multinucleated and dividing cells fluctuates over time, producing a wave-like alternation between biomass accumulation and reproductive output (Figure 9B). Under nutrient limitation, the oscillatory regime shifts toward smaller, rapidly dividing forms, reflecting adaptive plasticity in life-cycle dynamics. Thus, population structure emerges as an integrative property shaped by the feedback between individual cell cycles and environmental constraints.

Taken together, these findings support a systems-level model that links morphological transitions to metabolic state and population dynamics (Figure 9). In this model, each cell progresses along a continuum defined by three major phases: (i) biomass accumulation and nuclear multiplication, (ii) cytoplasmic reorganization and lipid redistribution, and (iii) cytokinesis and zoospore release. Feedbacks between these levels, mediated by nutrient availability and intercellular signaling, determine the balance between anabolic and dispersal phases. The transition thresholds are not fixed but context-dependent, allowing flexible adaptation to changing environmental conditions.

This integrated framework provides a quantitative foundation for modeling *T. aureum* culture dynamics. By incorporating parameters derived from image-based measurements, such as cell size distributions, nuclear number, and organelle ratios, computational models can simulate metabolic transitions and predict culture behavior under varying nutrient regimes. Such an approach is compatible with genome-scale metabolic and agent-based modeling strategies and offers potential applications for optimizing biotechnological processes, including lipid and omega-3 fatty acid production.

Overall, the integration of morphological, ultrastructural, and population-level observations establishes *T. aureum* ssp. *strugatskii* as a tractable model for studying the systems biology of protist growth and differentiation, continuing recent work on another species of thraustochytrid [17]. The life cycle of these organism exemplifies how self-organization and multilevel feedback regulate the balance between vegetative and reproductive states, providing a conceptual link between cell biology, ecology, and bioprocess engineering.

## 3. Materials and Methods

### 3.1. Isolate and Cell Culture

The isolate *T. aureum* ssp. *strugatskii* [11] was obtained from the Collective Center of the Institute of Cytology and Genetics SB RAS “Collection of Pluripotent Human and Mammalian Cell Cultures for Biological and Biomedical Research” (https://ckp.icgen.ru/cells/, accessed on 25 October 2025).

Cells were maintained in two culture media: (i) the FAND medium contained 5% (*v*/*v*) fetal bovine serum, 5% (*v*/*v*) Dulbecco’s Modified Eagle Medium (DMEM; prepared from dry concentrate dissolved in 17‰ artificial seawater), 0.05× non-essential amino acids (NEAA), and 1× Penicillin–Streptomycin (Thermo Fisher Scientific, New York, NY, USA) [11]; (ii) the YT medium consisted of 0.2% (by mass) yeast extract and 0.5% (by mass) tryptone (Angel, Yichang, China), following the modified formulation from [25]. In addition, glucose-enriched variants of these media (FANDg and YTg) were prepared by supplementing the base composition with an additional 2% (by mass) D(+)-Glucose (neoFroxx GmbH, Einhausen, Germany).

Cultures were incubated in 6-, 12-, or 24-well plates (Jet Bio-Filtration, Guangzhou, China) at room temperature (22–26 °C) in darkness. Media were renewed every two days. Nutrient concentration was controlled by serial dilution to prevent overgrowth or nutrient depletion.

To enrich samples with specific developmental stages, different media and timing schemes were used. For zoospore production, cells were seeded at a density of 2000 cells/cm^2^ in FAND medium and cultured for 24 h. The resulting zoospore-rich suspension was either immediately fixed for microscopy or transferred to culture plates containing substrates suitable for different imaging techniques. Coverslips were used for fluorescence microscopy (Celltreat, Ayer, MA, USA), Melinex polyester films (175 µm thickness; Agar Scientific, Rotherham, UK) for TEM, and silicon wafers for SEM (Agar Scientific, Rotherham, UK). After 10 min of attachment under visual control using an inverted microscope, cells were fixed according to the corresponding protocol.

To obtain vegetative cells and zoosporangia, cultures were grown in YT medium for two days under the same temperature and light conditions.

### 3.2. Light Microscopy and Time-Lapse Imaging

Light microscopy was used to monitor morphological changes and cell growth dynamics under different nutrient conditions. Protist cells were seeded into three wells of a 12-well culture plate at a density of 2000 cells/cm^2^. The FAND medium was diluted to concentrations of 1:1, 1:4, 1:8, and 1:16 to create nutrient gradients.

Time-lapse imaging was performed using two automated microscope–incubator systems: Cell-IQ (CM Technologies Oy, Tampere, Finland) and Image ExFluorer (LCI, Namyangju, Republic of Korea). Both platforms maintained stable environmental conditions and enabled automated acquisition of phase-contrast images at defined time intervals.

The Cell-IQ system was used for large-scale imaging, capturing multiple fields of view at 5 min intervals (TIFF format, image size 3923.57 × 2931.4 µm; 1392 × 1040 pixels). The Image ExFluorer system was employed for higher-resolution observation of individual cells and aggregates, with a frame interval of 2 min (ND2 format, image size 341.25 × 343.75 µm; 819 × 825 pixels).

Acquired image sequences were exported as 8-bit TIFF or AVI files and processed manually and semi-automatically. Cell division events were identified by an expert observer, and the number of daughter zoospores produced per division event was recorded. For cells that completed a full cycle, from motile zoospore attachment to the release of daughter zoospores, the projected cell area was measured using Fiji application (https://imagej.net/software/fiji/downloads, accessed on 25 October 2025) [26].

### 3.3. Fluorescence and Confocal Laser Scanning Microscopy

Confocal laser scanning microscopy was employed to visualize the cell wall, cytoplasm, and nuclei of *T. aureum* ssp. *strugatskii*. The culture was enriched for specific developmental stages prior to fixation. To obtain zoospores, cells were grown in FAND medium for 24 h. The medium enriched with free zoospores was transferred to culture wells containing glass coverslips, and zoospore attachment was monitored under an inverted microscope. After 10 min, attached cells were fixed. For enrichment of vegetative cells and zoosporangia, cultures were grown in YT medium for two days.

Fluorescent staining was performed using DAPI ($4^{\prime }$,6-diamidino-2-phenylindole, dihydrochloride, Sigma-Aldrich, St. Louis, MO, USA), CFW (Calcofluor White, Sigma-Aldrich, St. Louis, MO, USA), and PI (Propidium Iodide, Sigma-Aldrich, St. Louis, MO, USA). Reagents included 100% Triton X-100, 1× PBS, 32% paraformaldehyde (PFA). Working solutions were prepared as follows: 10% Triton X-100 in PBS (5 mL Triton X-100 per 45 mL PBS, incubated overnight); fixative solution (3.25 mL PBS, 0.5 mL 10% Triton X-100, 6.35 mL 32% PFA); and staining solution (PI, or CFW diluted 1:1000 in PBS). A 0.1% PBT wash buffer was prepared from 50 mL PBS and 0.5 mL of 10% Triton X-100.

The cell culture medium was removed and replaced with 0.5 mL of 1× PBS per well. After washing, 0.5 mL of the fixative solution was added and cells were fixed for 25 min. The fixative was removed, and the wells were washed three times with 0.1% PBT for 5 min each. Cells were then incubated for 30 min with the fluorescent staining solution (1:1000 dilution in PBS). After staining, the wells were washed three times with 1× PBS for 5 min each.

Slides were mounted in a DAPI-containing medium (glycerol with DAPCO) according to Rodig et al. [27].

Fluorescence imaging was performed at the Center of Collective Use for Microscopic Analysis of Biological Objects (https://ckp.icgen.ru/ckpmabo/, accessed on 25 October 2025), Institute of Cytology and Genetics SB RAS, using a Zeiss LSM 780 NLO confocal microscope (Zeiss, Oberkochen, Germany). Imaging was carried out in three spectral ranges: transmitted light (phase contrast) and two fluorescence channels with excitation wavelengths of 470 nm (DAPI emission, assigned blue pseudocolor) and 670 nm (PI emission, assigned red pseudocolor).

Z-stacks were acquired at resolutions of 800 × 800 or 512 × 512 pixels, with a pixel size of 0.082 or 0.040 µm and a voxel depth of 0.39 µm. Images were saved in OIR format at 16-bit depth.

### 3.4. Transmission Electron Microscopy

#### 3.4.1. Cell Preparation

For transmission electron microscopy, cells were prepared according to the protocol from [28]. Cells were cultured on Melinex polyester films (175 µm thickness; Agar Scientific, Rotherham, UK) placed in 12-well plates and fixed directly on the substrate in 2.5% glutaraldehyde prepared in culture medium for 15 min, followed by fixation in 2.5% glutaraldehyde in 0.1 M sodium cacodylate buffer (pH 7.3) for 1 h at room temperature. Samples were rinsed three times in the same buffer, post-fixed in 1% osmium tetroxide for 1 h, washed twice in double-distilled water, and incubated in 1% uranyl acetate for 12 h at 4 °C.

The fixed material was dehydrated through a graded ethanol series (30–100%, 10 min each step), followed by two changes in acetone (10 min each). Dehydrated samples were embedded in Epon 812 epoxy resin (Sigma, Saint Louis, MO, USA) and polymerized at 60 °C for 48 h. Throughout the fixation and embedding procedures, cells remained attached to the Melinex film. Areas with the highest cell density were identified on the polymerized resin, and 2 mm blocks were excised for sectioning.

Semi-thin sections (0.5 µm) were cut with a glass knife on an Ultracut ultramicrotome (Reichert, Depew, NY, USA), transferred onto glass slides in drops of distilled water, and dried on a histological plate at 60 °C. After drying, the sections were stained with 1% methylene blue in 1% aqueous sodium tetraborate, filtered through a 0.2 µm membrane. Semi-thin sections were examined under a light microscope (Axioscope, Zeiss, Oberkochen, Germany) to identify regions suitable for TEM analysis.

Ultrathin sections (50–70 nm) were cut parallel to the substrate plane using a diamond knife (Diatome, Nidau, Switzerland) on a Leica Ultracut UCT7 ultramicrotome (Leica, Vienna, Austria). Sections were examined using a JEM 1400 electron microscope (Jeol, Akishima, Japan) operated at 80 kV and equipped with a Veleta digital camera and iTEM 5.1 software (Olympus, Center Valley, PA, USA). Microscopy was performed at the Center of Collective Use for Microscopic Analysis of Biological Objects (https://ckp.icgen.ru/ckpmabo/, accessed on 25 October 2025), Institute of Cytology and Genetics SB RAS.

#### 3.4.2. Zoospore Preparation

For TEM analysis of *T. aureum* ssp. *strugatskii* cell and zoospore suspensions, samples were prepared following the protocol from [29]. Aliquots of 1 mL were subjected to prefixation in 2.5% glutaraldehyde prepared in culture medium for 15 min, followed by fixation in 2.5% glutaraldehyde in 0.1 M sodium cacodylate buffer (pH 7.2) for 1 h at room temperature. Samples were washed three times with the same buffer, post-fixed in 1% aqueous osmium tetroxide for 1 h at room temperature, washed with distilled water, and contrasted overnight in 1% aqueous uranyl acetate at 4 °C in the dark.

Dehydration was performed in a graded ethanol series (30, 50, 70, 96, and 100%) for 5, 5, 10, and two 10 min steps, respectively, followed by two rinses in acetone (10 min each). After each step, samples were gently resuspended using a plastic micropipette and centrifuged for 2 min at 200 g to preserve the cell pellet.

Embedding was performed by gradually replacing acetone with Epon 812 epoxy resin (Sigma, Saint Louis, MO, USA) using the following ratios of acetone to resin: 3:1, 1:1, 1:3, and pure resin, for 1.5, 1.5, and 2 h at each stage. Impregnation was carried out on a shaker with continuous mixing. Polymerization was completed in a thermostat at 60 °C for 48 h.

To identify areas enriched with zoospores, semi-thin sections (0.5 µm) were prepared from the polymerized blocks, stained with 1% methylene blue in 1% aqueous sodium tetraborate, and examined under a light microscope (Axioscope, Zeiss, Oberkochen, Germany).

Ultrathin sections (50–70 nm) were prepared with a diamond knife (Diatome, Nidau, Switzerland) on a Leica Ultracut UCT7 ultramicrotome (Leica, Vienna, Austria). Sections were examined using a JEM 1400 electron microscope (Jeol, Akishima, Japan) operated at 80 kV and equipped with a Veleta digital camera and iTEM 5.1 software (Olympus, Center Valley, PA, USA). All imaging was performed at the Center of Collective Use for Microscopic Analysis of Biological Objects (https://ckp.icgen.ru/ckpmabo/, accessed on 25 October 2025), Institute of Cytology and Genetics SB RAS.

#### 3.4.3. Morphometric Analysis of Organelles

Transmission electron microscopy (TEM) images were analyzed using open-source software for biological image processing, including Python libraries (scikit-image, OpenCV, NumPy, Pandas, scipy.stats), Napari (version 0.4.18), Microscopy Image Browser (version 2.91), and Fiji [30]. A total of 18 images of vegetative cells, 9 of the zoosporangium stage, 10 of undispersed zoospores, and 20 of free-floating zoospores were analyzed.

Image segmentation was performed semi-automatically. Mitochondria were identified using the Empanada plugin [31], while overall cell boundaries were delineated using Otsu thresholding followed by manual correction of segmentation errors. After automated segmentation, each image was manually refined and annotated into six morphological classes: nucleus, mitochondria, Golgi complex, paranuclear body, lipid droplets, and vacuoles.

Organellar and cellular areas were calculated using Python libraries (NumPy, OpenCV) based on binary masks. The length of inter-organelle contacts was measured in Microscopy Image Browser using the MCcale plugin (Plugins → Organelle Analysis → MCcale) with a minimum contact distance threshold of 200 nm and a maximum allowable gap of 5 pixels [32].

Quantitative comparisons between organelle groups and developmental stages were performed using the Mann–Whitney U test with multiple comparison corrections (Benjamini–Hochberg and Benjamini–Yekutieli procedures) implemented in the scipy.stats library. Differences were considered statistically significant at a *p*-value < 0.05.

#### 3.4.4. Image Processing and Three-Dimensional Model Rendering

Binary masks of organelles obtained from serial-section TEM images were used to reconstruct three-dimensional models of the zoosporangium and undispersed zoospores. Individual image tiles corresponding to different regions of the same cell were automatically stitched using Adobe Photoshop CS4 (https://www.adobe.com/products/photoshop.html, accessed on 25 October 2025). To improve stitching accuracy, histogram equalization and linear intensity transformation were applied to normalize image brightness and contrast.

Alignment of serial sections was performed using the TrakEM2 plugin for Fiji [33], followed by manual refinement and correction of non-linear distortions in Adobe Photoshop CS4. Missing data resulting from incomplete series of sections were represented as blank slices in the reconstructed stacks.

Three-dimensional surface rendering and visualization were carried out in Microscopy Image Browser [34] using the organelle masks generated from segmented images.

### 3.5. Field-Emission Scanning Electron Microscopy (FE-SEM)

For FE-SEM analysis of *T. aureum* ssp. *strugatskii* cells, sterile silicon chips (5 × 5 mm; Agar Scientific, Rotherham, UK) were used as substrates. Each chip was coated with a drop of poly-*L*-lysine solution (1 µg/mL in distilled water) for 30 min, rinsed with water, and air dried to produce a hydrophilic surface.

Approximately 10–20 µL of cell suspension was placed onto the coated chips positioned at the bottom of microtube chambers. The samples were briefly centrifuged (1000× *g*, 1 min) to promote attachment and then fixed in 2.5% glutaraldehyde prepared in 0.1 M sodium cacodylate buffer containing 5 mM MgCl_2_ for 10 min at room temperature. Following fixation, chips were rinsed with the same buffer and post-fixed in 1% osmium tetroxide (OsO_4_) for 10 min at room temperature. Samples were then washed with distilled water and stained with 1% aqueous uranyl acetate for 10 min.

Dehydration was carried out through a graded ethanol series (10, 30, 50, 70, 90, and 100%), followed by critical-point drying from liquid CO_2_ using a Leica EM CPD030 (Leica Microsystems, Wetzlar, Germany).

The dried specimens were mounted on aluminum stubs and coated with a conductive layer of gold–palladium alloy. Imaging was performed at an accelerating voltage of 30 kV using a Hitachi S-5200 field-emission scanning electron microscope (Hitachi, Tokyo, Japan) at the Center for Collective Use “Nanostructures,” Institute of Semiconductor Physics SB RAS (https://www.isp.nsc.ru/, accessed on 25 October 2025).

#### Data Visualization

Statistical analysis and data visualization were performed using Python (v. 3.11.9) and open-source scientific computing libraries. Data preprocessing and numerical operations were conducted with NumPy (v. 2.3.3) and pandas (v. 2.3.3), which enable efficient handling of tabular and time-series data. Graphical representations and statistical plots were generated using matplotlib (v. 3.10.6) and seaborn (v. 0.13.2), providing clear visual summaries of experimental results.

### 3.6. Analysis of Cell Growth Dynamics

Data analysis was performed using the Python programming language (version 3.11.9) with the following libraries: pandas (version 2.3.3) for reading and processing CSV files, numpy (version 2.3.3) for numerical computations and array operations, scipy (version 1.15.2) for curve fitting and growth model approximation, matplotlib (version 3.10.6) for data visualization and plotting, and seaborn (version 0.13.2) for boxplot generation and statistical visualization.

Cell growth data were obtained from three experimental conditions (20–21 cells per condition): undiluted medium, 1:4 dilution, and 1:16 dilution of FAND medium (Appendix A). The primary data consisted of time series of cell projected area measurements at multiple time points (in minutes). Assuming spherical cell morphology, the projected area (*S*) was converted to cell diameter (*d*) using the formula: $d=2\sqrt{\frac{S}{\pi }}$, where *S* is the measured cell area. This transformation allowed conversion from projected area to a more intuitive size metric (diameter) suitable for growth dynamics analysis.

To describe cell growth mathematically, several empirical models were tested for curve fitting:Exponential model: $d\left(t\right)=d_{0}\cdot e^{k\cdot t}$, where $d_{0}$ is the initial diameter and *k* is the exponential growth rate.Logistic model: $d\left(t\right)=\frac{K}{1+e^{-r\cdot (t-t_{mid})}}$, where *K* is the maximum diameter (growth capacity), *r* is the growth rate parameter, and $t_{mid}$ is the time to reach half of the maximum size.Linear model: d(t)=a·t+b, where *a* is the linear growth rate and *b* is the initial size.

Model selection was based on the mean coefficient of determination ($R^{2}$) across all cells. The comparison yielded the following results: linear model, mean $R^{2}=0.940$ (n=62); exponential model, mean $R^{2}=0.951$ (n=62); logistic model, mean $R^{2}=0.962$ (n=62). The logistic model provided the best overall fit and was selected for subsequent processing. For consistency, all individual cell trajectories were fitted using the same logistic model framework.

The following quantitative metrics were derived for each cell (Appendix A):Maximum diameter ($d_{max}$): the largest observed cell size.Initial diameter ($d_{0}$): the size at the first observation point.Final diameter ($d_{final}$): the size at the last observation point.Lifetime (*T*): duration between first and last observations (in hours).Mean growth rate (*v*): average rate of diameter increase, calculated as $v=\frac{d_{final}-d_{0}}{T}$, where $d_{final}$ and $d_{0}$ are measured in μm and *T* in hours.Doubling time ($T_{double}$): estimated time required for the cell diameter to double relative to its initial value.

The logistic model parameters were interpreted as follows (Appendix A): *K* (μm), the maximum achievable cell size under given conditions; *r* ($h^{-1}$), the rate at which the cell approaches the maximum size; and $t_{mid}$ (h), the time point at which the growth rate reaches its maximum (inflection point). Comparison of these parameters across dilution conditions allowed quantitative evaluation of how nutrient concentration affects growth behavior.

To illustrate average growth dynamics and within-group variability (Appendix A):All individual logistic models were extrapolated to the same temporal range (from the start of observation to the maximum experimental duration).The mean growth curve for each condition was computed as the arithmetic mean of fitted values at each time point.The variability (±1 standard deviation) was visualized as a shaded region around the mean curve.

For visualization and comparison, boxplots and descriptive statistics (mean and standard deviation) were generated (Appendix A).

This analytical workflow enabled both the assessment of average growth trajectories and the estimation of intra-group variability under different nutrient conditions. Overall, the logistic model captured cell growth in *T. aureum* ssp. *strugatskii* with high precision, providing a reliable basis for quantitative comparisons among experimental conditions.

## 4. Conclusions

This study presents the first integrative reconstruction of the life cycle of *T. aureum* ssp. *strugatskii* using a combination of multimodal imaging and quantitative morphometric analysis. By unifying data across molecular, cellular, and population scales, we demonstrate that the organism’s development follows a self-regulating, cyclic pattern that balances biomass growth with reproductive output. Our observations show that synchronized nuclear division and organelle reorganization are key drivers of morphogenesis, while population-level oscillations between vegetative and reproductive phases emerge from collective feedbacks within the culture. These findings support the concept of multilevel coordination between metabolic flux, cell architecture, and ecological dynamics. The systems-level framework developed in this study outlines how coordinated growth, nuclear replication, and cell-cycle transitions contribute to the stability of *T. aureum* ssp. *strugatskii* populations under different environmental conditions. Although preliminary, this conceptual model may support the development of more refined genome-scale and agent-based simulations, ultimately aiding the rational optimization of cultivation regimes for lipid and omega-3 fatty acid production. Taken together, our results indicate that *T. aureum* ssp. *strugatskii* possesses several features that make it a promising experimental system for investigating self-organization and life-cycle regulation in unicellular eukaryotes though it is not yet fully established. At the same time, additional multi-omics datasets and functional experiments will be required to confirm its status relative to existing protist model organisms and to further validate the generality of the proposed framework. 

## Figures and Tables

**Figure 1 ijms-26-11302-f001:**
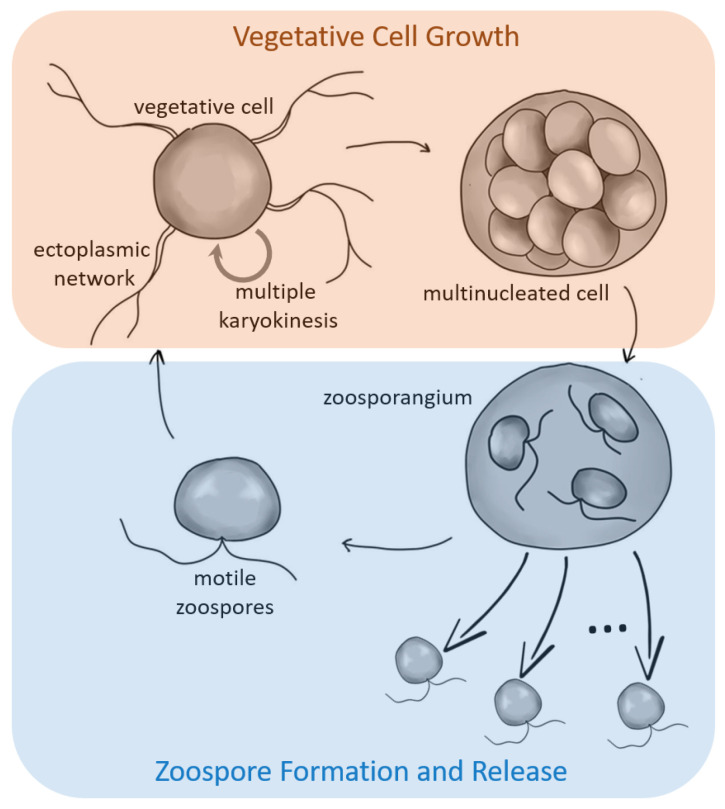
Overview of the life cycle of the marine protist *T. aureum* ssp. *strugatskii*. Schematic representation of the main developmental stages, including motile zoospores, vegetative cells, multinucleated (zoosporangium-like) stages, and zoospore release. Arrows indicate transitions between phases, highlighting two distinct functional cycles: a growth phase (biomass accumulation and synchronized nuclear division) and a dispersal phase (zoospore formation and release).

**Figure 2 ijms-26-11302-f002:**
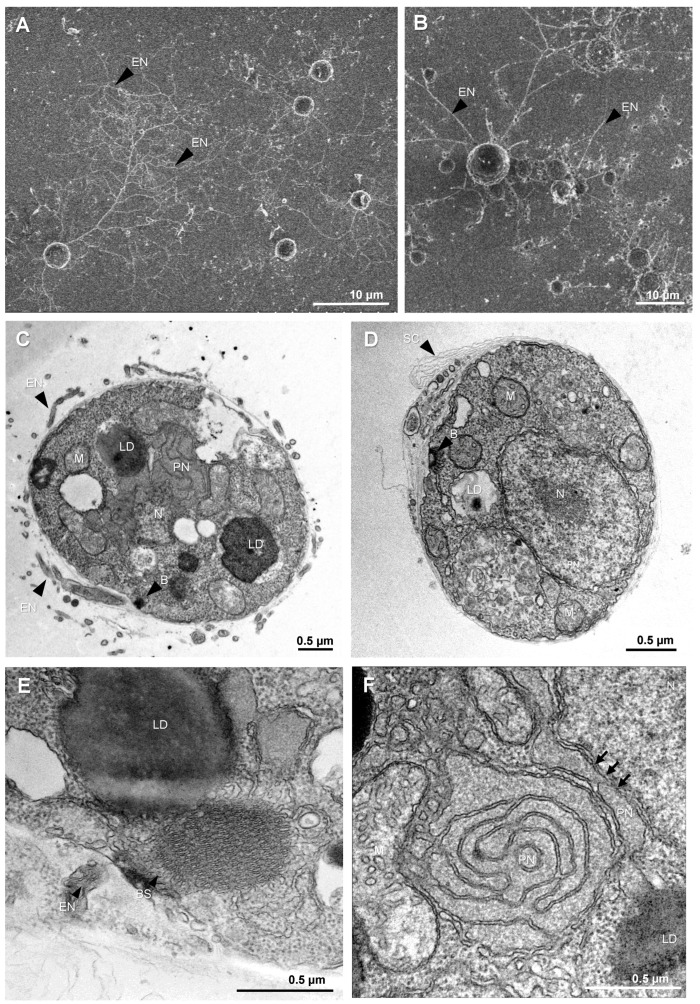
Vegetative cell morphology and ultrastructure in *T. aureum* ssp. *strugatskii*. (**A**) SEM image of a synchronous culture showing early vegetative cells. Round cells produce ectoplasmic extensions that form an ectoplasmic net (EN). (**B**) SEM image of an asynchronous culture showing cells of different sizes with extensive ectoplasmic material attached to the substrate surface. (**C**,**D**) TEM images of vegetative cells showing the nucleus (N), mitochondria (M), paranuclear body (PN), and large lipid droplets (LD) in the cytoplasm, as well as the electron-dense material of the bothrosome (B) at the cell periphery. The cell surface is covered by multiple layers of scales (SC), and fragments of the ectoplasmic net (EN) surround the cell. (**E**) TEM image of a vegetative cell fragment lacking large lipid droplets and bothrosomes, shown at higher magnification. (**F**) TEM image of the paranuclear body (PN) at high magnification, located between the nucleus and a mitochondrion; arrows indicate the contacts between the PN and the nuclear envelope. Abbreviations: EN, ectoplasmic net; N, nucleus; M, mitochondrion; G, Golgi apparatus; PN, paranuclear body; LD, lipid droplet; B, bothrosome; SC, scales.

**Figure 3 ijms-26-11302-f003:**
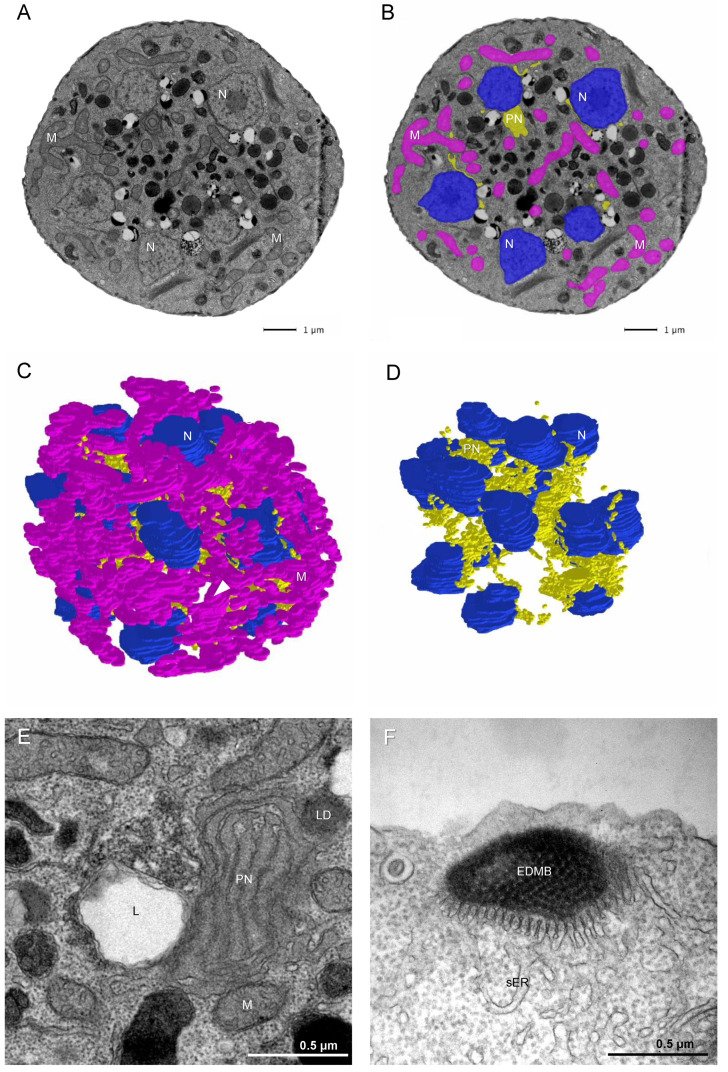
Ultrastructural organization of the multinucleated vegetative cell of *T. aureum* ssp. *strugatskii*. (**A**) TEM image of an ultrathin section of a multinucleated vegetative cell. (**B**) The same image with color overlays indicating major organelles: blue, nucleus (N); pink, mitochondria (M); yellow, paranuclear body (PN). (**C**) Three-dimensional reconstruction of the ultrastructure of a multinucleated vegetative cell from serial ultrathin sections. (**D**) The same reconstruction as in (**C**), with mitochondria omitted for clarity. (**E**) The paranuclear body (PN) occupies a substantial portion of the central volume of the multinucleated cell. (**F**) Bothrosome (B) observed at the zoosporangium stage. Abbreviations: N, nucleus; M, mitochondrion; PN, paranuclear body; L, lysosome; LD, lipid droplet; sER, smooth endoplasmic reticulum; EDMB, electron-dense material of the bothrosome.

**Figure 4 ijms-26-11302-f004:**
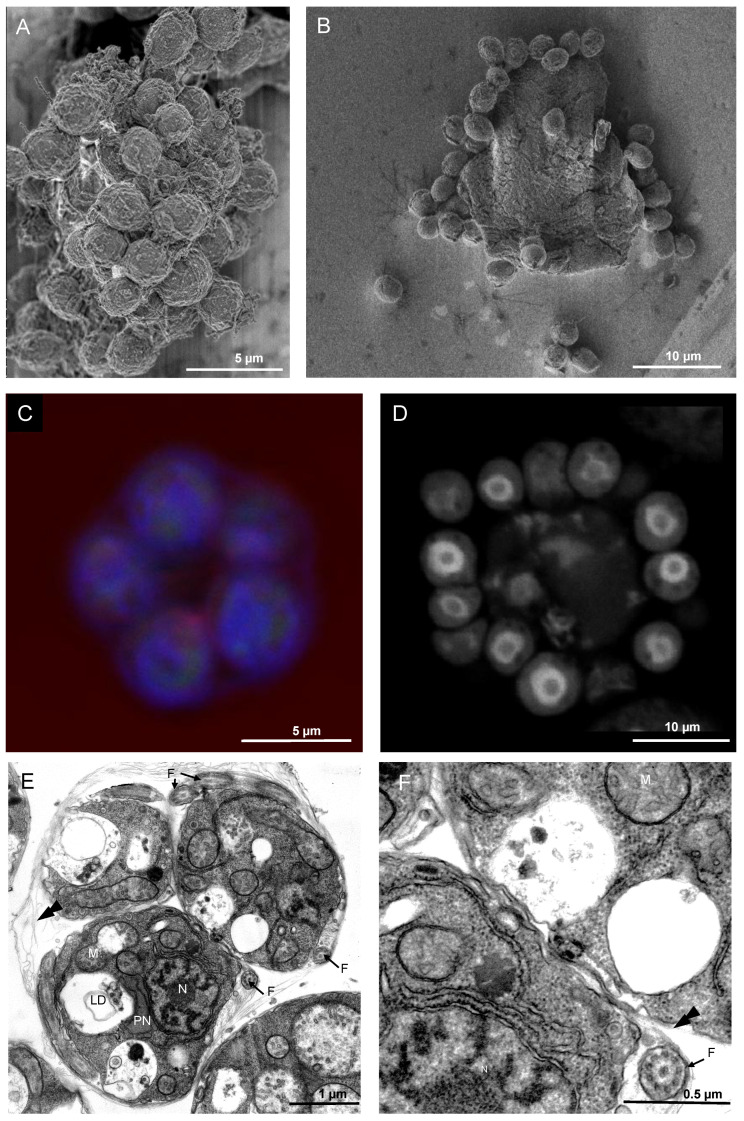
Morphology of cytokinesis in multinucleated vegetative cells of *T. aureum* ssp. *strugatskii*. (**A**) SEM image showing the result of cell division at the late stages of karyokinesis, represented by a group of unseparated zoospores. (**B**) SEM image showing a group of released zoospores located on the surface of the disrupted plasma membrane of the former zoosporangium. (**C**) Laser scanning microscopy image showing the early stages of karyokinesis: a group of zoospores beneath the plasma membrane, with a central region free of cellular material. The central optical section is shown. Blue, DAPI; red, PI. (**D**) Laser scanning microscopy image showing the late stages of karyokinesis: a group of released zoospores with a central region containing a residual body of undivided cytoplasm (indicated by the letter T). The central optical section is shown; individual zoospores are marked by arrows. PI staining. (**E**) TEM image showing undispersed zoospores of *T. aureum* ssp. *strugatskii*. General view of a zoospore group. (**F**) TEM cross-section of an undispersed zoospore showing a flagellum (F) at high magnification; the typical 9+2 microtubule arrangement is visible in the cross section. Abbreviations: N, nucleus; M, mitochondrion; PN, paranuclear body; F, flagellum; double arrowheads, scales.

**Figure 5 ijms-26-11302-f005:**
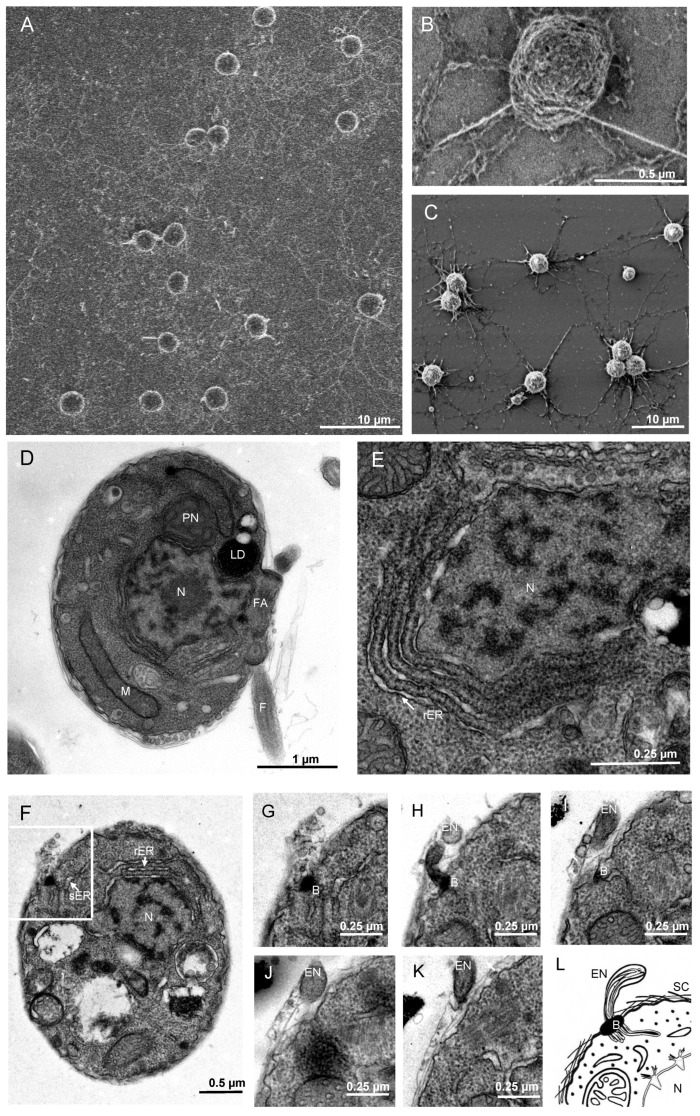
*T. aureum* ssp. *strugatskii* at the zoospore stage. (**A**) SEM image showing zoospores of *T. aureum* ssp. *strugatskii* after attachment. (**B**) SEM image of an individual zoospore bearing two flagella (F). (**C**) Growing zoospores lose their flagella and begin to form an ectoplasmic network (EN) becoming a young vegetative cell. (**D**) TEM image showing the ultrastructure of a zoospore. (**E**) TEM (image showing parallel lamellae of the rough endoplasmic reticulum (rER) at high magnification. (**F**) TEM image showing the formation of the bothrosome (B) and the associated ectoplasmic network (EN). (**G**–**K**) Serial TEM sections showing the formation of the bothrosome (B) and the associated ectoplasmic network (EN). (**L**) Schematic representation of ectoplasmic network formation through the bothrosome. Abbreviations: N, nucleus; M, mitochondrion; PN, paranuclear body; LD, lipid droplet; FA, flagellar apparatus; F, flagellum; EN, ectoplasmic network; sER, smooth endoplasmic reticulum; rER, rough endoplasmic reticulum; B, bothrosome.

**Figure 6 ijms-26-11302-f006:**
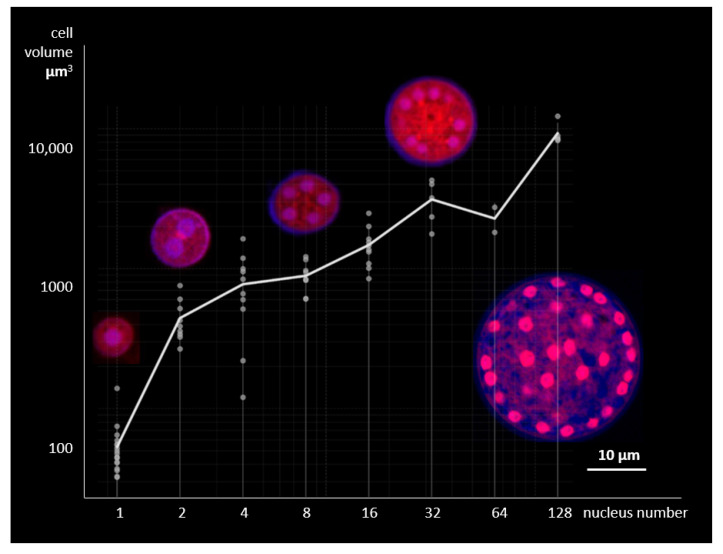
Relationship between cell volume and nuclear number in laboratory culture of *Thraustochytrium aureum* ssp. *strugatskii* grown in standard FAND medium (Appendix A). Scatterplot shows the correlation between total cell volume and the number of nuclei (both axes in logarithmic scale). Representative optical sections corresponding to different stages of karyokinesis are shown above the plot. Fluorescent staining: Propidium Iodide (PI, red), $4^{\prime }$,6-diamidino-2-phenylindole dihydrochloride (DAPI, blue), and Calcofluor White (CFW, blue). Each data point represents a single cell reconstructed from confocal laser scanning microscopy *z*-stacks, with quantitative measurements of cell volume and nuclear count.

**Figure 7 ijms-26-11302-f007:**
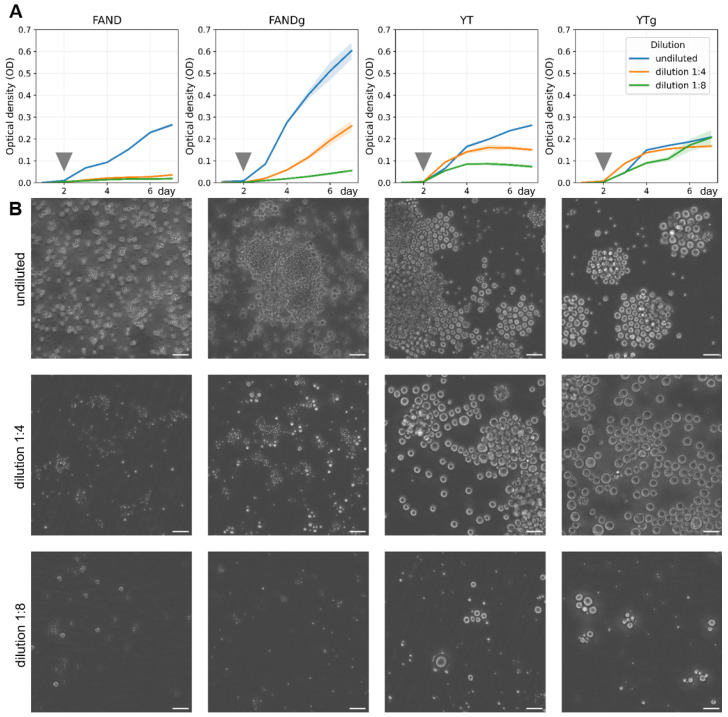
Growth dynamics of *Thraustochytrium aureum* ssp. *strugatskii* under different culture conditions. (**A**) Changes in optical density (OD) over time for four media: FAND, FAND supplemented with glucose (FANDg), YT, and YT supplemented with glucose (YTg). The graph shows average OD dynamics for three dilution levels: undiluted (blue), 1:4 dilution (orange), and 1:8 dilution (green). Solid lines represent the mean OD values, and shaded areas indicate ±SD based on biological replicates (n=3). Measurements were performed daily for each medium–dilution combination (Appendix A). Gray arrows mark the time points corresponding to the population snapshots shown in (**B**). (**B**) Representative phase-contrast images of *T. aureum* ssp. *strugatskii* populations in the respective media and dilution conditions on the second day of cultivation. Differences in OD trajectories reflect distinct growth rates and nutrient utilization patterns among media. Scale bar = 35 µm.

**Figure 8 ijms-26-11302-f008:**
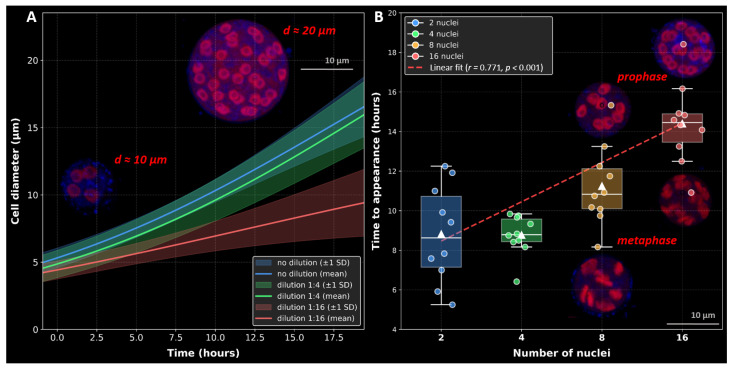
Growth dynamics and timing of nuclear multiplication in *T. aureum* ssp. *strugatskii*. (**A**) Average cell growth curves under three dilution conditions of FAND medium: undiluted (blue), 1:4 dilution (green), and 1:16 dilution (red). Each curve represents the mean logistic growth model fitted to all cells within the corresponding dilution group (Appendix A, Appendix A). Shaded areas denote ±1 standard deviation. Insets show a four-nucleated cell (diameter, 10 µm) and a 32-nucleated cell (diameter, 20 µm), maximum intensity projection of the LSM-stack. Fluorescent staining: Propidium Iodide (PI, red), $4^{\prime }$,6-diamidino-2-phenylindole dihydrochloride (DAPI, blue), and Calcofluor White (CFW, blue). (**B**) Relationship between the number of nuclei per cell and the time required for cell appearance. Boxplots show medians, interquartile ranges, and whiskers indicating the full data range; diamond symbols represent means, and solid circles indicate individual cell observations (Appendix A). The red dashed line shows a linear regression fit (r=0.771, p<0.001). Groups are color-coded: blue (2 nuclei, n=10), green (4 nuclei, n=10), orange (8 nuclei, n=10), and red (16 nuclei, n=10). ANOVA indicated significant group differences (F=18.56, p<0.001). Pairwise post hoc comparisons (Bonferroni correction) revealed significant differences between 2 and 16 nuclei (p<0.001), 4 and 8 nuclei (p<0.01), 4 and 16 nuclei (p<0.001), and 8 and 16 nuclei (p<0.01). A strong positive correlation was observed between nuclear number and time to appearance (r=0.771, p<0.001). Additional insets show 8- and 16-nucleated cells in prophase and anaphase stages, indicating synchronous karyokinesis, maximum intensity projection of the LSM-stack. Fluorescent staining: PI (red), DAPI (blue), and CFW (blue).

**Figure 9 ijms-26-11302-f009:**
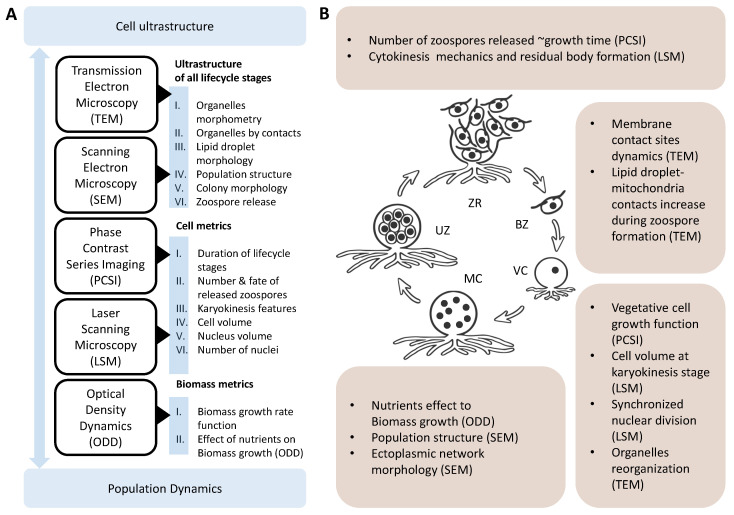
Composition of microscopy methods used for the integrative reconstruction of the life cycle of *T. aureum* ssp. *strugatskii* and the main classes of results obtained through quantitative morphometric analysis. (**A**) Summary of imaging techniques and the principal data types they provide. Each method corresponds to a specific level of biological organization, covering a range from the ultrastructure of individual organelles to the behavior of single cells and the dynamics of population composition. (**B**) Novel structural patterns and quantitative dependencies identified across the life cycle stages of *T. aureum* ssp. *strugatskii*. Abbreviations: VC, vegetative cell (juvenile sporangium); MC, multinucleated cell (mature sporangium); UZ, undispersed zoospores; ZR, zoospore release stage; BZ, biflagellate zoospore.

## Data Availability

The original contributions presented in this study are included in the article/Appendix A. Further inquiries can be directed to the corresponding author.

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
