# Peer review of "Integrative Study of the Life Cycle in the Marine Protist Thraustochytrium aureum ssp. strugatskii"

_ijms, 2025, doi:10.3390/ijms262311302_

Round 1

Reviewer 1 Report

Comments and Suggestions for Authors

This article holds considerable innovative significance but requires revisions to address several minor issues:

  1. Introduction Section The background on the biotechnological applications of Labyrinthulea organisms is overly general. It fails to clarify the differences in metabolic characteristics and cultivation advantages between T. aureum ssp. strugatskii and other well-studied Thraustochytrium species (e.g., Schizochytrium sp., Aurantiochytrium sp.), resulting in an insufficiently highlighted rationale for this research. Additionally, the statement "However, most existing models rely on a limited number of input parameters and do not explicitly account for the multi-level organization of living systems, particularly the dynamic structure of cell populations" requires more illustrative examples.
  2. Materials and Methods Section For instance, in the description of culture media: "Cells were maintained in two culture media: (i) the FAND medium contained 5% fetal bovine serum, 5% Dulbecco’s Modified Eagle Medium (DMEM; prepared from dry concentrate at 17‰ artificial seawater), 0.05× non-essential amino acids (NEAA), and 1× Penicillin–Streptomycin (Thermo Fisher Scientific, USA) [11]; (ii) the YT medium consisted of 0.2% yeast extract and 0.5% tryptone (Angel, China), following the modified formulation from [25]." The specific concentration units (e.g., mass/volume percentage, volume/volume percentage) for the "%" values should be explicitly defined.
  3. Conclusion and Discussion The claim that "this species is established as a tractable model organism for studying the systems biology of protists" lacks sufficient experimental support. Further evidence is needed to demonstrate its unique advantages compared to existing model organisms (e.g., Plasmodium falciparum, Schizochytrium limacinum). Additionally, the limitations of current research are not fully discussed—for example, whether model-based calculations can fully represent real-life biological states has not been adequately addressed. Would you like me to refine the English expression further or expand on specific parts (such as adding example supplements for the introduction or clarifying concentration unit descriptions)?

Reviewer 2 Report

Comments and Suggestions for Authors

The paper presents the results of a profound study of morphology, ultrastructure and life cycle of a marine Labyrinthulean Thraustochytrium aureum ssp. strugatskii. This is an interesting emerging model object perspective from the point of cell biology, life cycles of protists, and biotechnology. The authors perform a thorough description of morphology and ultrastructure at different stages of the life cycle of this protist. I do not have criticisms on the results obtained by the authors; the study is conducted on a very high technical level. However, I have two main suggestions for the improvement of the paper.

The first one concerns the structure of the paper. As the Results and Discussion section preceds Methods, it is a bit difficult to follow findings from the point of non-specialist. What I was missing, was a clear summariziang explanation of the life cycle stages sequence (preferably with illustrations) and an explanation of which methods were applied to which stages (i.e. Fig. 9 would be appropriate in the beginning). Descriptions of different stages and results obtained by different methods should be more structured than they are now. I.e. (hypothetically) 1.1 Zoospore formation 1.1.1 Light microscopy 1.1.2 TEM ... 1.1.N. Analysis and conclusions. In my opinion, this could make the paper easier to follow.  By the way, the life cycle scheme presented in the beginning (Fig. 1) implies a haplodiploid cycle (there are 2N and N), but I have not seen ay details on recombination and sexual process in this cycle further. If this is not what you mean, the letters are a bit confusing.

Another suggestion, although not directly related to the subject of the paper, needs to be expressed, and this concerns the taxonomic status of the isolate. In my opinion the use of "subspecies" concept in the protistan nomenclature makes it less clear and chaotic. There is even no clear understanding in the majority of the clades, what a species is, and designating subspecies increases this ambiguity. I understand that the revision of the group is long overdue, and there are multiple criteria for distinction of strains that probably do not always agree and hardly work well, but especially in this situation increasing the number of taxonomic ranks does not improve the situation. I read also the original publication where the strain was described; I am not sure I understand the reasons for establishing a subspecies and not a separate species.

Comments on the Quality of English Language

The language looked clear to me, but I am not a native speaker. Some parts of the text seem to be wordy and may be shortened. For example, "these observations demonstrate that the vegetative cell of T. aureum ssp. strugatskii is a structurally complex system that combines typical eukaryotic organization with distinctive labyrinthulean features." (lines 154-156) seems to be an obvious statement; I am not sure it is necessary in the context of the paper.
